# Reduction in SOCE and Associated Aggregation in Platelets from Mice with Platelet-Specific Deletion of Orai1

**DOI:** 10.3390/cells11203225

**Published:** 2022-10-14

**Authors:** Linlin Yang, Roger Ottenheijm, Paul Worley, Marc Freichel, Juan E. Camacho Londoño

**Affiliations:** 1Pharmakologisches Institut, Ruprecht-Karls-Universität Heidelberg, INF 366, 69120 Heidelberg, Germany; 2DZHK (German Centre for Cardiovascular Research), Partner Site Heidelberg/Mannheim, 69120 Heidelberg, Germany; 3The Solomon H. Snyder Department of Neuroscience, School of Medicine, Johns Hopkins University, Baltimore, MD 21205, USA

**Keywords:** calcium signalling, store operated Ca^2+^ entry (SOCE), receptor-operated Ca^2+^ entry (ROCE), platelet aggregation, Orai1 proteins, platelet specific knockout mice

## Abstract

Calcium signalling in platelets through store operated Ca^2+^ entry (SOCE) or receptor-operated Ca^2+^ entry (ROCE) mechanisms is crucial for platelet activation and function. Orai1 proteins have been implicated in platelet’s SOCE. In this study we evaluated the contribution of Orai1 proteins to these processes using washed platelets from adult mice from both genders with platelet-specific deletion of the *Orai1* gene (Orai1^flox/flox^; Pf4-Cre termed as Orai1^Plt-KO^) since mice with ubiquitous Orai1 deficiency show early lethality. Platelet aggregation as well as Ca^2+^ entry and release were measured in vitro following stimulation with collagen, collagen related peptide (CRP), thromboxane A2 analogue U46619, thrombin, ADP and the sarco/endoplasmic reticulum Ca^2+^-ATPase (SERCA) inhibitor thapsigargin, respectively. SOCE and aggregation induced by Thapsigargin up to a concentration of 0.3 µM was abrogated in Orai1-deficient platelets. Receptor-operated Ca^2+^-entry and/or platelet aggregation induced by CRP, U46619 or thrombin were partially affected by Orai1 deletion depending on the gender. In contrast, ADP-, collagen- and CRP-induced aggregation was comparable in Orai1^Plt-KO^ platelets and control cells over the entire concentration range. Our results reinforce the indispensability of Orai1 proteins for SOCE in murine platelets, contribute to understand its role in agonist-dependent signalling and emphasize the importance to analyse platelets from both genders.

## 1. Introduction

Calcium as a second messenger regulates signalling pathways crucial for platelet activation and function, including secretion and thrombus formation. In non-excitable cells, like platelets, Ca^2+^ signalling is amplified and sustained through two main mechanisms: The store operated Ca^2+^ entry (SOCE), which occurs after depletion of the intracellular Ca^2+^ stores [1,2,3], and receptor-operated Ca^2+^ entry (ROCE) [4,5,6,7]. In platelets, both Ca^2+^ entry mechanisms occur and have been investigated in murine and human platelets. Particularly in platelets the release of Ca^2+^ comes from the dense tubular system (mediated by PLC-dependent synthesis of 1,4,5-triphosphate -IP_3_-), the analogue of the endoplasmic reticulum, but not exclusively since release from acidic organelles, including lysosomes associated with the second messenger nicotinic acid adenine dinucleotide phosphate, was discussed in human platelets [8]. 

Platelets express several receptors regulating their function. Many of them can be grouped into G-protein coupled receptors (GPCR) and immunoreceptor tyrosine based activation motif (ITAM)-linked receptors. These groups of receptors, which can be activated by different physiological agonists such as collagen, thromboxane A2 (TxA2), thrombin or ADP, retain clear differences in signalling pathways including signatures of Ca^2+^ mobilization from the endoplasmic reticulum [9,10]. Platelets express several receptors for collagen, including the integrin α2β1 having a major role in adhesion and platelet anchoring, and the GPVI, which is found as complex with the homodimeric Fc receptor γ-chain and mediates signalling, platelet activation and thrombi stabilization [9,11,12,13,14,15,16]. Thromboxane A2 (TxA2), thrombin and ADP activate different GPCRs coupled to different G-proteins, including G_12/13_, G_q_ and/or G_i_; thromboxane A2 (TxA2) or its analogue U46619 activate TP receptors, ADP activates P2Y receptors including P2Y_12_ and P2Y_1_, the latter being responsible for ADP-evoked Ca^2+^ transients. Thrombin activates protease-activated receptors (PAR), being PAR3 and PAR4 expressed in mice and PAR1 and PAR4 in humans [17]. A common feature of both groups of receptors is the downstream activation of phospholipase C (PLC) proteins (PLCγ2 after GPVI activation or PLCβ after G_q_-coupled receptor activation), which by formation of IP_3_ leads to depletion of Ca^2+^ stores and eventually induces SOCE where Orai1 proteins are involved [9,18,19,20,21].

Orai1 proteins were identified as a ion conducting pore of highly Ca^2+^ selective SOCE channels [22,23,24,25] and Orai1, Orai2 and Orai3 have been termed [26,27]. However, the input of each Orai isoform to endogenous SOCE can variates in primary cells as shown in subtypes of lymphocytes [28]. The role of Orai1 proteins in murine and human platelets and its contribution for SOCE has been described and reviewed [10,21,29,30,31]. In human platelets, especial attention on the regulation of Orai1 by STIM proteins has been developed, which act as sensors of Ca^2+^ levels inside the stores and after Ca^2+^ store depletion STIM proteins get in close proximity to the plasma membrane where they trigger activation of Orai1 [32,33,34,35].

The role of Orai1 in murine platelets has been analysed in two models; one model is based on the expression of a mutated, inactive form of Orai1 (Orai^R93W^), which is homologous to a mutation found in immunodeficiency patients [36]. In the second model bone marrow chimeric mice, i.e., lethally irradiated wild-type mice that received a transplant of Orai1^−/−^ or control bone marrow cells, were studied and showed protection of pathological thrombus formation [37]. Platelets from Orai^R93W^ mice presented reduced SOCE, impaired agonist-induced increase in intracellular Ca^2+^ concentration and αIIbβ3 activation, but unchanged platelet aggregation [36]. Similarly, the analysis of Orai1^−/−^ platelets bone marrow chimeric mice showed reduced SOCE, but different to the Orai^R93W^ mice some impairment in Ca^2+^ signalling and platelet aggregation were reported, especially regarding GPVI signalling [37]. Remarkably, the same group showed that Orai1 deletion in platelets did not completely abolish SOCE and some residual portion was attributed to TRPC6 channels [38,39]. Additionally, Orai1^−/−^ bone marrow chimeric mice served to validate the specificity of thromboprotective action of 2-aminoethyl diphenylborinate (2-APB) [30] supporting the concept of the use of SOCE blockers as anti-platelet drugs.

In our current study, we analysed platelets from adult mice with platelet-specific Orai1 deletion from both genders to determine in vitro the contribution of Orai1 proteins to platelet aggregation and Ca^2+^ signalling. Our results reinforce that SOCE essentially relies on the presence of Orai1 proteins in murine platelets, and they further contribute to understand Orai1’s role in GPVI-dependent signalling and emphasize the importance to analyse platelets from both genders. 

## 2. Materials and Methods

### 2.1. Animal Experiments

Blood sampling and animal handling were approved and performed according to the regulations of the Regierungspräsidium Karlsruhe and the University of Heidelberg conform to the guidelines from Directive 2010/63/EU of the European Parliament on the protection of animals used for scientific purposes. Blood was obtained by submandibular bleeding as described [40] but using an 18G 11/2” needle under terminal Ketamine/Xylazine anaesthesia (Xylazine-hydrochloride and Ketamine-hydrochloride final dose 16 and 120 mg/kg body weight, respectively; solved in sterile NaCl 0.9% and volume of 5 µL/g body weight; intraperitoneal injection). Mice were maintained under specified pathogen-free conditions in the animal facility (IBF) of the Heidelberg Medical Faculty. Mice were housed in a 12 h light–dark cycle, with a relative humidity between 56–60%, a 15-times air change per hour and room temperature of 22 ± 2 °C. They were kept in conventional cages type II or type II long provided with animal bedding LTE E-001 (ABBEDD, Germany) and tissue papers as enrichment. Standard autoclaved food (Rod 16 or 18, Altromin, Lage, Germany) and autoclaved water were available to consume ad libitum. Investigators were blinded towards genotype and treatment of the mice. After data analysis and confirmation of the genotype from tail biopsies the genotype was released to the investigators. On each experimental day both genotypes were analysed in parallel and the starting genotype for blood isolation were changed among days to avoid bias due to time differences of blood isolation. To generate a platelet-specific Orai1 knock out mouse model (Orai1^Plt-KO^) we mated Orai1^flox/flox^ mice [41] with C57BL/6-Tg(Pf4-icre)Q3Rsko/J (Stock Nr. 008535) from The Jackson Laboratory (Bar Harbor, MA, USA) to obtain Orai1^flox/flox^; Pf4-Cre (termed as Orai1^Plt-KO^) and Pf4-Cre negative Orai1^flox/flox^ mice termed as Orai1^Plt-Ctrl^ as controls. Male and female mice between 9–21-week-old were used. 

### 2.2. Reagents

All reagents were from analytical grade and their sources are listed here. Tri-sodium citrate dihydrate (A2403, AppliChem, Darmstadt, Germany), citric acid monohydrate (10655102, Fischer Scientific, Loughborough, UK), glucose monohydrate (108342, Merck, Darmstadt, Germany), NaCl (A3597, AppliChem, Darmstadt, Germany), KCl (31248, Sigma-Aldrich, Taufkirchen, Germany), Hepes (9105.4, Carl Roth, Karlsruhe, Germany), MgCl_2_-6H_2_O (2189.2, Carl Roth, Karlsruhe, Germany), NaHCO_3_ (10152780, Fischer Scientific, Loughborough, UK), Na_2_HPO_4_-2H_2_O (A3567, AppliChem, Darmstadt, Germany), bovine serum albumin—BSA albumin fraction V (8076.2, Carl Roth, Karlsruhe, Germany), prostacyclin (P6188, Sigma-Aldrich, Taufkirche, Germany), heparin-Na (Pzn-7833909, Ratiopharm, Ulm, Germany), apyrase (from potato grade VII, A6535, Sigma-Aldrich, Taufkirche, Germany), EGTA (E4378, Sigma-Aldrich, Taufkirche, Germany), Fura-2 LR/AM (344911, Sigma-Aldrich, Taufkirche, Germany), EDTA disodium salt (A2937, AppliChem, Darmstadt, Germany), Collagen Reagens HORM^®^ Suspension (1130630, Takeda, Linz, Austria), collagen related peptide (CRP-XL, AB_281019M, CambCol Laboratories, Ely, UK), bovine thrombin (T4648, Sigma-Aldrich, Taufkirche, Germany), ADP (A2754, Sigma-Aldrich, St. Luis, MO, USA), fibrinogen (F3879, Sigma-Aldrich, St. Luis, MO, USA), U46619 (Cay16450, Cayman Chemica, Ann Arbor, MI, USA) and thapsigargin (T9033, Sigma-Aldrich, Taufkirche, Germany).

### 2.3. Isolation of Washed Murine Platelets and Platelet Aggregation

Washed platelets were isolated similarly as previously reported [42]. Briefly, blood was pooled from 3 to 5 mice of the same genotype and gender. Blood was collected in tubes containing acid-citrate-dextrose (tri-sodium citrate dihydrate 2.5%, citric acid 1.4%, glucose 2%) solution and then mixed (0.75 volumes) with a modified Tyrode’s buffer (137 mM NaCl, 2 mM KCl, 5 mM Hepes, 4.5 mM glucose, 1 mM MgCl_2_, 12 mM NaHCO_3_ and 0.3 mM NaHPO_4_, pH 7.4 and 37 °C) supplemented with 0.35% BSA and 2 mM CaCl_2_. Platelet rich plasma was obtained by centrifugation at 2300 G (~10 s/mL final volume) in a Heraeus Multifuge X3R (Thermo Scientific) at 24 °C. Platelet pellets were resuspended and washed by two centrifugation steps at 1075 G/5 min. The first centrifugation was done in the presence of prostacyclin (0.5 µM final concentration) and the second in the presence of both, prostacyclin and heparin (10 U/mL). Subsequently, cell pellets were washed at the same conditions without inhibitors. Platelet numbers were determined, adjusted to 3 × 10^8^ platelets/mL and cells were incubated for 30 min at 37 °C in the presence of 0.02 U/mL apyrase before aggregation experiments. Aggregation was determined at 37 °C using two APACT 4S Plus (Diasys, Holzheim, Germany) aggregometers, allowing measuring 8 samples in parallel. Tyrode’s buffer served as reference value of 100% light transmission. A 190 µL platelet suspension was used and stimulated with corresponding agonists in a volume of 10 µL after 90 s of basal recording. In case of ADP-induced aggregation, measurements were done in the presence of 70 µg/mL fibrinogen (F3879, Sigma-Aldrich, Taufkirche, Germany) added (4 µL) shortly before starting the measurements. For additional experiments (See Appendix A) and to prevent the secondary activation by TxA2, washed platelets were incubated with 1 mM Aspirin (A2093, Sigma-Aldrich, Taufkirche, Germany) for 10 min before stimulation and in some cases additionally 10 U/mL apyrase were added before measurements to prevent the effects of secreted ADP.

For Ca^2+^ imaging the Tyrode’s buffer (see above) was used, but initial addition of Ca^2+^ was omitted to have a nominally free Ca^2+^ solution (nominal 0 Ca^2+^). Slightly different to the cell isolation for aggregation the first washing step was done directly with buffer supplemented with both, prostacyclin and heparin. After, cells were incubated with Fura-2 LR/AM for 30 min at room temperature and protected from light. Platelets were finally washed and re-suspended in Tyrode’s buffer (0 nominal Ca^2+^). Platelet concentration was adjusted to 1 × 10^8^ platelets/mL and cells were incubated at room temperature for 30 min in the presence of 0.02 U/mL apyrase before experiments.

### 2.4. Ca^2+^ Imaging of Mouse Platelets

Experiments were done as previously reported [42] using a spectrofluorometer (F7000 Hitashi). Shortly, platelets were stimulated at 37 °C with continuous stirring in the absence of extracellular Ca^2+^. Where indicated EGTA (1 mM) was added to remove extracellular Ca^2+^. To determine the agonist-induced Ca^2+^-release, platelets were stimulated in the absence of extracellular Ca^2+^. After the release event, CaCl_2_ (3 mM) was added to determine the Ca^2+^ entry. Fura-2 LR/AM was excited alternately at 340 and 380 nm, and fluorescence emission was detected at 510 nm. Fluorescence signals were corrected for autofluorescence determined from unloaded cells. For calibration of fluorescence ratios, EDTA (4 mM final concentration) was added after the measurements to get values in the absence of Ca^2+^ and after stable signal Triton (0.1% final concentration) was added to obtain fluorescence values in saturating Ca^2+^ conditions. Fluorescence ratios, R (340/380), were calibrated in terms of the intracellular concentration of Ca^2+^ with the following equation: [Ca_2+_]i = Kd (So/Ss) (R − Rmin)/(Rmax − R), where So and Ss are the fluorescence at 380 nm in the absence of Ca^2+^ and in saturating Ca^2+^, respectively [43], and Kd (the dissociation constant) is 290 nM [44]. Area under the curve (AUC) of the Ca^2+^ trace above basal levels was calculated between 90 and 360 s for Ca^2+^ release and between 360 and 540 s for Ca^2+^ entry phases. Ca^2+^ peak was measured as maximal Ca^2+^ concentration within 30 s after either agonist application or Ca^2+^-addition. To determine the required extracellular Ca^2+^ concentration to evoke Orai1 dependent Ca^2+^ entry we tested both 0.3 mM and 3 mM Ca^2+^ after addition after thapsigargin. Therefore, a concentration of about 2 mM extracellular free-Ca^2+^ is expected in the presence of 1 mM EGTA (Calculated using an online tool—https://somapp.ucdmc.ucdavis.edu/pharmacology/bers/maxchelator/CaEGTA-TS.htm—based on [45] (accessed on 18 September 2022) and slightly lower ~1.2 mM considering the presence of BSA in the buffer used. 

### 2.5. Statistical Analysis

Data were analysed and processed with Excel 2010 (Microsoft), Graphpad Prism 8.0 and OriginPro 2019G ver. 9.6.0.172 (OriginLab). Values are shown as mean ± standard deviation (SD), except for original traces of Ca^2+^ imaging as indicated in the corresponding figure legends. For dose–response curves a non-linear regression was done for graphical representation. Statistical significances were assumed when *p* < 0.05. *p*-values are depicted as * *p* < 0.05; ** *p* < 0.01, *** *p* < 0.001. Statistical analysis was performed using the appropriate tests (unpaired Student’s *t*-test, Mann-Whitney U test or Kruskal-Wallis followed by Dunn’s test) after normality test (Shapiro-Wilk normality test).

## 3. Results

Deletion of Orai1 proteins leads to perinatal lethality in mice that can be rescued only partially by backcrossing to an outbred mouse strain [46]. To overcome this obstacle, we generated a platelet-specific Orai1 deficient mouse line by mating Orai1^flox/flox^ mice [41] with Pf4-Cre mice. We isolated platelets from Pf4-Cre positive male or female mice (Orai1^Plt-KO^) and compared them with corresponding litter matched control animals (Orai1^Plt-Ctrl^) in terms of in vitro platelet aggregation and Ca^2+^-signalling to determine the contribution of Orai1 proteins after stimulation with different platelet agonists at different concentration (Collagen, collagen related peptide (CRP), Thromboxane A2 analogue U46619, thrombin and ADP) as well as with the sarco/endoplasmic reticulum Ca^2+^-ATPase (SERCA) inhibitor, thapsigargin. We analysed platelet aggregation in the presence of 2 mM total extracellular Ca^2+^ concentration; taking into account that our buffer contains albumin, it is assumed that the free Ca^2+^ is about ~1.2 mM similarly as the physiological free plasma Ca^2+^ concentration. In the case of intracellular Ca^2+^ measurements we performed a stepwise Ca^2+^ re-addition protocol having two extracellular Ca^2+^ concentrations (>0.1 and ~2 mM). This approach proved to be useful to analyse Orai proteins in other cellular systems [47]; it improves the chance of identifying an involvement of Orai1 in Ca^2+^ entry since it is possible that very high extracellular concentrations mask the function of Orai1 by recruiting other channels or due to a saturating Ca^2+^ signal. 

### 3.1. Analysis of the Role of Orai1 Proteins in Platelet Function Downstream of the Glycoprotein VI (GPVI) Receptor and the PLCγ Axis

Orai1 proteins have been implicated in murine platelet function and Ca^2+^-signalling downstream collagen receptors, especially after activation of Glycoprotein VI (GPVI) receptors by CRP [37,38]. To analyse this signalling pathway we compared the in vitro platelet aggregation from Orai1^Plt-KO^ and Orai1^Plt-Ctrl^ washed platelets from female and male mice after stimulation with collagen (Figure 1A–D). Collagen at a concentration of 3µ g/mL provokes an intermediate maximal aggregation response that was comparable between both genotypes in platelets from female (Figure 1A) or male (Figure 1B) mice. A collagen concentration of 10 µg/mL evoked near maximal aggregation responses indistinguishable between Orai1^Plt-KO^ and Orai1^Plt-Ctrl^ male platelets (Figure 1C). Dose response curves for maximal aggregation and area under the aggregation curve is shown in Figure 1 and did not reveal an effect of Orai1 deletion. To further dissect the GPVI-dependent signalling in platelets, which mediates Ca^2+^-signalling and platelet activation through PLCγ [9,11], we stimulated platelets with CRP and analysed platelet aggregation (Figure 1E,F) and Ca^2+^ signalling (Figure 1G). After stimulation with 0.3 µg/mL CRP we observed variability in the maximal aggregation response, which did not differ in platelets from female Orai1^Plt-KO^ mice; additional CRP concentrations showed similar responses between both genotypes (Figure 1E). Additionally, in male mice the aggregation responses were comparable between the two genotypes at all CRP concentrations tested (Figure 1F). The maximal aggregation after 1.0 µg/mL CRP was significantly different, but the absolute difference was minimal (Orai1^Plt-Ctrl^ = 90.7% ± 1.61% and Orai1^Plt-KO^ = 88.2 ± 2.64%) and it was not different in terms of area under the aggregation curve (Figure 1F). To further dissect the aggregation response, we performed experiments in the presence of acetyl salicylic acid (ASA) and high apyrase concentration to inhibit secondary activation by produced TxA2 and secreted ADP, respectively. However, under these conditions CRP-induced aggregation was similar between Orai1-deficient platelets and corresponding control ones (Appendix A). Finally, we tested the effect of Orai1 deletion on the CRP-induced Ca^2+^-signalling in platelets from male and female mice (Figure 1G). Both, the CRP-induced Ca^2+^ release in the absence of extracellular Ca^2+^ as well as the Ca^2+^-entry evoked by the re-addition of 1 mM extracellular Ca^2+^ were similar between Orai1^Plt-KO^ and Orai1^Plt-Ctrl^ platelets from males, but both parameters were reduced in Orai1-deficient platelets isolated from female mice compared to the corresponding controls. Here, is to note that the responses from male and female platelets are different in terms of the levels of raise in Ca^2+^ concentration. 

### 3.2. Analysis of ORAI1’s Role in Platelet Function Downstream of G-Protein-Coupled Receptors and PLCβ Axis

Platelet function is also regulated by physiological agonists like thrombin, thromboxane A2 (TxA2) or ADP. Those agonists activate G-protein coupled receptors (GPCRs) such as protease-activated receptors (PARs), P_2_Y receptors and TxA2 receptor (TP), respectively. These receptors are coupled to different G-proteins enabling regulation of Ca^2+^-signalling though mechanisms including PLCβ-downstream signalling [9,48]. Consequently, we tested if Orai1 deletion has an effect on in vitro platelet aggregation and Ca^2+^-signalling. Platelet aggregation induced by 1 µM of the TxA2 receptor agonist analogue U46619 was normal in platelets from Orai1^Plt-KO^ female mice (Figure 2A) but significantly reduced in platelets from Orai1^Plt-KO^ male mice (Figure 2B). Higher U46619 concentrations used in male platelets provoked similar responses between platelets from Orai1^Plt-KO^ and from Orai1^Plt-Ctrl^ mice (Figure 2C,D). We also aimed to analyse U46619-induced aggregation independent of secondary activation by produced TxA2 and secreted ADP. However, the presence of apyrase abolished completely U46619-induced aggregation in both genotypes independent of U46619 concentration (data not shown); therefore, we did experiments in the presence of ASA alone and observed a similar reduction in both genotypes (Appendix A). U46619 1 µM provoked a fast Ca^2+^ release in platelets from male and female Orai1^Plt-KO^ mice comparable to the corresponding controls (Figure 2E,F). However, the initial Ca^2+^ entry represented by the peak in Ca^2+^ rise evoked after the addition of extracellular Ca^2+^ was reduced in platelets from both genders of Orai1^Plt-KO^ mice, being significant in males (Figure 2E). 

We further analysed the thrombin-induced platelet aggregation and found that concentrations of 0.1 U/mL (Figure 3A,B) and 0.3 U/mL (Figure 3C,D) triggered similar responses in Orai1-deficient platelets and control cells from both genders. Furthermore, under conditions inhibiting secondary activation by produced TxA2 and secreted ADP, platelet aggregation was similar between platelets from both genotypes (Appendix A). In contrast, the Ca^2+^-entry evoked by 0.1 U/mL in male platelets of Orai1^Plt-KO^ mice was significantly reduced (Figure 3E). Ca^2+^ signalling after thrombin stimulation appeared normal in platelets from Orai1^Plt-KO^ female mice (Figure 3F). 

The ADP-induced platelet aggregation was also analysed; 1 µM ADP induced aggregation in platelets from Orai1^Plt-KO^ female and male mice was unaltered as compared to control cells (Figure 4A,B), as well as aggregation induced by any other ADP concentration tested in male platelets (Figure 4C,D). In an additional set of experiments, we further tested the effect of epinephrine. Epinephrine, which activates α2A receptors in platelets, is considered a weak platelet agonist that despite contributing to thrombus stability alone does not induces aggregation in isolated platelets from different species [49,50,51]; therefore, we did experiments in the presence collagen at a concentration that alone does not induces platelet aggregation. No differences in epinephrine/collagen aggregation between control and Orai1-deficient platelets were detected (Appendix A). Finally, we explored the effect of arachidonic acid, a precursor of TxA2, on aggregation and similar responses with a high variability were observed between both genotypes (Appendix A). 

### 3.3. Reduction in Store Operated Calcium Entry (SOCE) and Associated Platelet Aggregation

In murine platelets, either the global deletion of Orai1 proteins or the expression of a mutated inactive form of Orai1 (Orai1^R93W^) affected the Store Operated Calcium Entry (SOCE) [19,37], which is a known important mechanism involved in cellular Ca^2+^ signalling in non-excitable cells. We then evaluated the consequence of platelet-restricted Orai1 deletion on SOCE in platelets from both genders. For this, we used the sarco/endoplasmic reticulum (ER) Ca^2+^-ATPase (SERCA) inhibitor, thapsigargin, which leads to passive depletion of intracellular Ca^2+^ stores. Thapsigargin applied at a concentration of 0.1 µM induced a near maximal platelet aggregation in Orai1^Plt-Ctrl^ male mice, which was completely abolished in platelet from both genders of Orai1^Plt-KO^ mice (Figure 5A,B). A 3-fold higher thapsigargin concentration (0.3 µM) was also not able to trigger aggregation in platelets from Orai1^Plt-KO^ male mice (Figure 5C) and it was impaired in females (Appendix A). Further higher thapsigargin concentrations showed the impairment of platelets from Orai1^Plt-KO^ mice to normally aggregate under SERCA blockage (Figure 5C and Appendix A). Analysis of thapsigargin-induced (0.3 µM) aggregation in the presence of ASA and high apyrase concentration was just slightly affected in Orai1^Plt-Ctrl^ male platelets (Appendix A), indicating that the abolished response observed in Orai1^Plt-KO^ platelets is rather not produced by potential defects on secondary activation produced TxA2 and secreted ADP. Analysis of intracellular Ca^2+^ signalling revealed similar Ca^2+^ release from the intracellular Ca^2+^ stores, but Ca^2+^ entry was largely reduced under physiological extracellular Ca^2+^ concentrations in platelets from both genders of Orai1^Plt-KO^ mice (Figure 5E,F). 

## 4. Discussion

Store operated Ca^2+^ entry (SOCE) and receptor-operated Ca^2+^ entry (ROCE) mechanisms are central for platelet activation and function. Orai1 proteins have been implicated in platelet’s SOCE and in regulation of arterial thrombosis [52]. To further understand the involvement of Orai1 proteins in platelet function, we analysed Orai1-deficient platelets from adult male and female mice regarding platelet aggregation and Ca^2+^ signalling. We used physiological agonist at different concentrations such as collagen, collagen related peptide (CRP), thromboxane A2 analogue U46619, thrombin, ADP and the SERCA inhibitor, thapsigargin, and determined the impact of Orai1 deletion in mouse platelets. Because Orai1 global deficient mice die early and to specifically analyse the function of Orai1 in adult platelets, we generated a platelet-specific knockout mouse line (Orai1^Plt-KO^). Orai1-deficient platelets had strongly reduced thapsigargin-induced SOCE resulting in abrogated aggregation, independent of secondary activation produced TxA2 and secreted ADP. Ca^2+^-entry and platelet aggregation induced by U46619 or thrombin were partially affected by Orai1 deletion depending on the gender. ADP, collagen and CRP aggregation from Orai1-deficient platelets was comparable with control cells, and only CRP-induced Ca^2+^-signalling was affected in female platelets at the tested concentrations. 

In previous studies the function of Orai1 in murine platelets was analysed in two models; one model based on the expression of a mutated, inactive form of Orai1 (Orai^R93W^) [36], and other model is based in the use of platelets from bone marrow chimeric Orai1^−/−^ mice that show protection of pathological thrombus formation [37]. Platelets from Orai^R93W^ mice presented reduced SOCE (produced by thapsigargin), impaired agonist-induced increase in intracellular Ca^2+^ concentration (stimulated by PAR4 peptide or convulxin, a GPVI agonist) and normal aggregation triggered by PAR4 peptide, collagen, ADP and convulxin [36]. Similarly, the analysis by Braun and colleagues [37] of Orai1^−/−^ platelets reported strongly reduced SOCE produced by thapsigargin and showed a reduced increase in CRP-induced Ca^2+^ transients in the absence of extracellular Ca^2+^; in addition, apparently Ca^2+^ transients in the presence of extracellular Ca^2+^ were reduced after stimulation with thrombin, CRP or ADP, but platelet aggregation after ADP and thrombin were normal. In contrast to the Orai^R93W^ model, Orai1^−/−^ platelets showed reduced GPVI-mediated platelet aggregation when evoked by low dose CRP (0.05 µg/mL) or collagen (1 µg/mL), respectively, but platelet aggregation was not affected at higher concentrations of either agonist [37]. In another later study from the same group, platelet aggregation was measured in Orai1^−/−^ platelets and it was shown to be normal after thrombin, ADP, thromboxane A2 analogue U46619 and collagen stimulation, reduced after low dose CRP stimulation and completely abolished after thapsigargin stimulation [39]. When we compared our findings with those summarized above in terms of in vitro aggregation, we found similar results to those from platelets isolated from Orai^R93W^ mice [36], since we did not observe significantly reduced aggregation after collagen or GPVI stimulation (Bergmeier and collaborators reported to be normal after convulxin, we used CRP), which, is in contrast to what it was reported from Orai1^−/−^ platelets that is reduced response to those agonists [37,39]. Regarding TxA2 evoked platelet activation, we report that U46619 (0.6 µM)-induced aggregation is reduced in Orai1^−/−^ platelets from male mice, similarly as found by Chen and co-workers who used reported reduced aggregation in Orai1^−/−^ platelets after stimulation with 0.5 µM U46619 but not at higher (2 µM) concentrations [39]. When the Orai1^−/−^ studies are compared with ours here, we tested here several concentrations and quantified the response from several independent preparations and from both gender of the animal used. A common point among all reports and us in terms of aggregation is the preserved ADP and thrombin responses [36,37,39], and abolished thapsigargin response [39]. From the Ca^2+^ measurements the thapsigargin induced SOCE was strongly reduced, but not completely abolished, in male and female platelets here as well as in platelets from Orai1^−/−^ or Orai^R93W^ mice [36,37,39]. We further observed reduced Ca^2+^ entry after thrombin (in male platelets) similarly to others [39]. In comparison to others, we observed the reduced responses for U46619 (males) and normal after CRP (males), which discern from those described in Orai1^−/−^ platelets [37,39]; this can be partially explained by differences in the concentrations used, but also by gender because we observed reduced CRP-induced SOCE only in platelets from female mice. Further analysis of Orai1^−/−^ platelets by other showed reduction in Ca^2+^ transients evoked no only by CRP or thrombin but also by combined thrombin/CRP stimulation and determined the presence of residual SOCE [38], which was partially attributed to TRPC6 channels since platelets from double Orai1^−/−^/TRPC6^−/−^ mice had stronger thapsigargin-induced SOCE compared to single Orai1^−/−^ platelets [39]. 

The dissection of SOCE in murine platelets revealed how in vitro this Ca^2+^-entry is affected by conditions where coagulation components are or not present. From the study of Orai1 deficient platelets Orai1-mediated SOCE specifically regulates (in vitro) GPVI-induced integrin activation and degranulation, and thrombus formation under flow [37]. Further analysis with the same murine model by Gilio et al. [38] showed that deficiency of Orai1 inhibits GPVI-mediated phosphatidylserine (PS) exposure and thrombus formation associated with reduced Ca^2+^ signalling; however, when the same parameters were observed under conditions allowing coagulation, PS exposure and thrombus formation were preserved in Orai1-deficient platelets, and very interesting a modest reduction in Ca^2+^ signalling was observed. Even more, the deficiency in Orai1 did not affect GPVI-Induced PS exposure and prothrombinase activity in washed platelets when thrombin is present, but it affected thrombin generation after GPVI stimulation. Altogether and presuming that in washed platelets the coagulation components from plasma are mainly reduced, we assume that our observations can be partly evoked by inhibition of the procoagulant platelet formation and further studies combining approaches like those from the studies mentioned above can be used in the future.

In human platelets corresponding studies evaluating the contribution of individual Orai subtypes to SOCE is difficult due to the lack of inhibitors with sufficient specificity (i.e., small molecules or antibodies) of the molecules involved in its regulation like STIM or Orai proteins. However, attempts to study the contribution of SOCE in isolated human platelets have been performed. In one the study using washed human platelets the SOCE evoked by stimulation of thrombin and thapsigargin, respectively, was not affected by an anti-STIM antibody, despite that this antibody had an inhibitory effect on collagen-induced aggregation and thrombus formation on collagen coated capillary under flow [53]. In another study the effect of different SOCE inhibitors, including the CRAC blocker GSK 7975A [54], was evaluated in human platelets after stimulation either with thrombin or convulxin; the specificity of the inhibitors was evaluated using Orai1-deficient platelets and a thrombus formation assay but unfortunately not in the Ca^2+^ signalling setup [30]. 

On the other hand, differences in the observations between our work and the work published by others can be in part attributed to the mouse model type used and the approach used to inactive Orai1 proteins. Two main differences between our approach and those using the Orai1^−/−^ model [37,38,39] are: First, the Orai1^−/−^ model is based in transplantation of bone marrow into irradiated wild-type mice, and second, the targeting used was gene-trapping, which most probably resulted in a hypomorphic Orai1 allele [24] in comparison to the murine model produced by targeted homologous deletion of exons 2 and 3 [46]. We used here a model of specific deletion in platelets using the expression of Cre under the Pf4 promotor, as others did it to analyse other important SOCE regulator (STIM1) in mouse platelets [55]. The Orai1 inhibition in adult murine platelets by the expression of a dominant-negative mutant (Orai1^R93W^) [36] differed from ours since a dominant negative Orai1 subtype may affect all channel entities consisting of Orai1 proteins, and the composition of the native channel. In addition, the mutant not only deactivates Orai1 channels, but alters the function of its signalosome too [21]. Importantly, we analysed separately platelets from mice of both genders. Differences between genders in platelet responses to agonists and platelet counts have been reported [56,57] and looking forward into gender differences in pre-clinical studies is a pivotal component to design better pharmacological options for patients of both genders. Unfortunately, regarding gender we cannot really compare our work with the previous work since this was not clearly reported [37,38,39,55]. 

Another influencing factor that need to be considered is that the expression of all Orai1, Orai2 and Orai3 has been reported in both, murine and human platelets [36,37,58]; therefore, it is feasible that deletion of Orai1 can be compensated by other isoforms making difficult to isolate the Orai1 contribution. Despite a clear differentiation of some molecular mechanisms in platelet signalling, in vitro, in the physiological context activation of relevant pathways in platelets by agonist such as collagen, thromboxane A2 (TxA2), thrombin or ADP triggers downstream parallel activation of PLCβ or PLCγ2 isoforms; PLC activation is capable to recruit both SOCE and ROCE dependent Ca^2+^ entry mechanisms, by further IP3 and diacylglycerol (DAG) formation and or PIP2 depletion [9,18,20,21]; in addition, one has to consider that channels composed of Orai1 proteins can also be activated by non-store operated mode [59]. Therefore, the putative roles determined by in vitro assays like here could be occurring differently in the complex physiological context with activation of multiple receptors and signalling pathways. 

Finally, our results support the concept of Orai1 involvement into SOCE mechanism in murine platelets, contribute to understand its role in agonist-dependent signalling and pointed out the importance to analyse separately platelets from both genders. 

## Figures and Tables

**Figure 1 cells-11-03225-f001:**
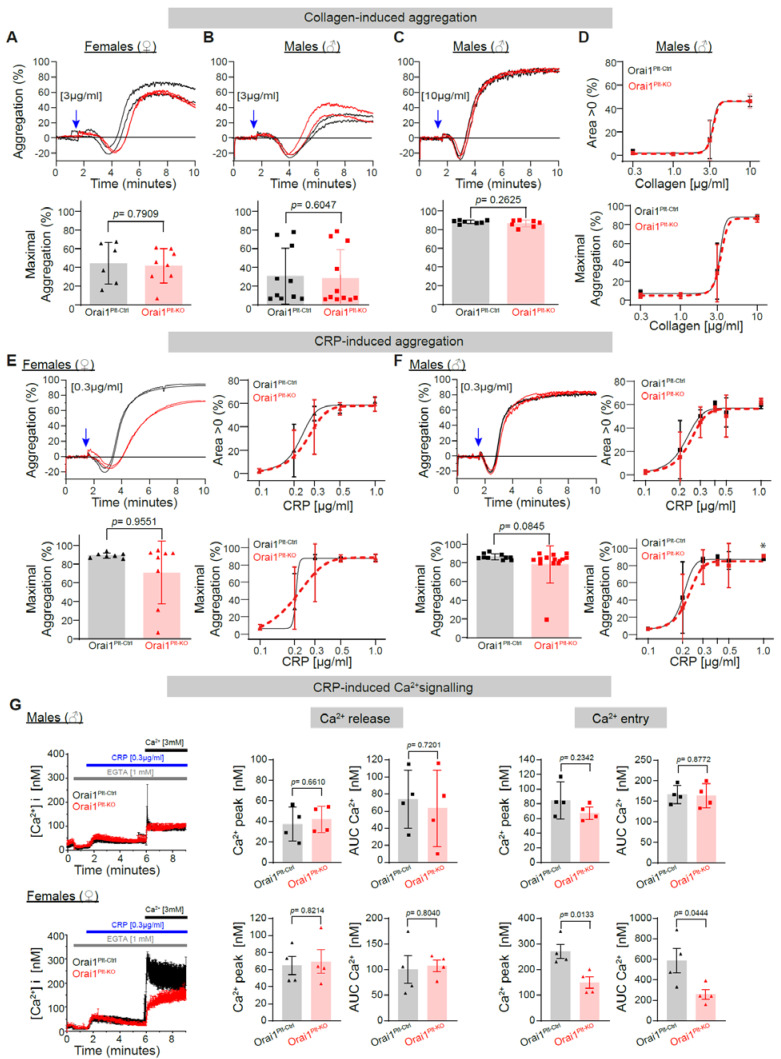
Collagen- and CRP-induced platelet aggregation and Ca^2+^ signalling in Orai1 deficient platelets. Collagen-induced platelet aggregation in washed platelets from Orai1^Plt-Ctrl^ (black) and Orai1^Plt-KO^ (red) mice was analysed (**A**–**D**). Representative original aggregation traces (**upper panels**) and analysis of maximal aggregation after stimulation with 3 µg/mL collagen of platelets from female (**A**) or male (**B**) mice; similarly, analysis of platelets from male mice stimulated with 10 µg/mL collagen are depicted (**C**). Dose-response curves for the area under the aggregation curve (**upper panel**) and the maximal aggregation after collagen stimulation of male platelets are shown (**D**). Each dot in (**A**–**C**) corresponds to one independent platelet preparation. In (**D**) numbers of independent platelet preparations per collagen concentration were: 0.3 µg/mL *n* = 3, 1 µg/mL *n* = 4, 3 µg/mL *n* = 10/11 and 10 µg/mL *n* = 7. CRP-induced platelets aggregation was analysed in platelet from female (**E**) and male mice (**F**). Left panels show the aggregation response after stimulation with 0.3 µg/mL CRP and right panels depict the dose-response curves for the area under the aggregation curve (**upper panels**) and the maximal aggregation (**lower panels**) after CRP stimulation. Numbers of independent preparations per CRP concentration for females (**E**) were: 0.1 µg/mL = 6/8, 0.2 µg/mL = 5/6, 0.3 µg/mL = 7/8, 0.5 µg/mL = 6/8 and 1 µg/mL = 6/8, for males (**F**) were: 0.1 µg/mL = 7/8, 0.2 µg/mL = 9, 0.3 µg/mL = 10/11, 0.4 µg/mL = 3, 0.5 µg/mL = 9/10 and 1 µg/mL = 8/9. Blue arrows indicate the agonist stimulation time point. In (**G**) average Ca^2+^ traces of Fura-2 LR/AM loaded platelets after CRP stimulation of platelets from male (**upper panels**) and female (**lower panels**) mice are shown; corresponding statistical analysis of maximal Ca^2+^ peak and area under the curve (AUC) of Ca^2+^ release and Ca^2+^ entry (*n* = 4) upon stimulation of platelets from male mice with 0.3 µg/mL CRP are depicted. Error bars indicate SD. Original Ca^2+^ traces are represented as mean ± SEM for better visualization. *p*-values or * *p* < 0.05 were calculated according to the unpaired Student’s *t*-test (Panels **A**, **C** and **G**) or Mann-Whitney U test (**B**, **E** and **F**) after Shapiro-Wilk normality test. CRP: Collagen related peptide.

**Figure 2 cells-11-03225-f002:**
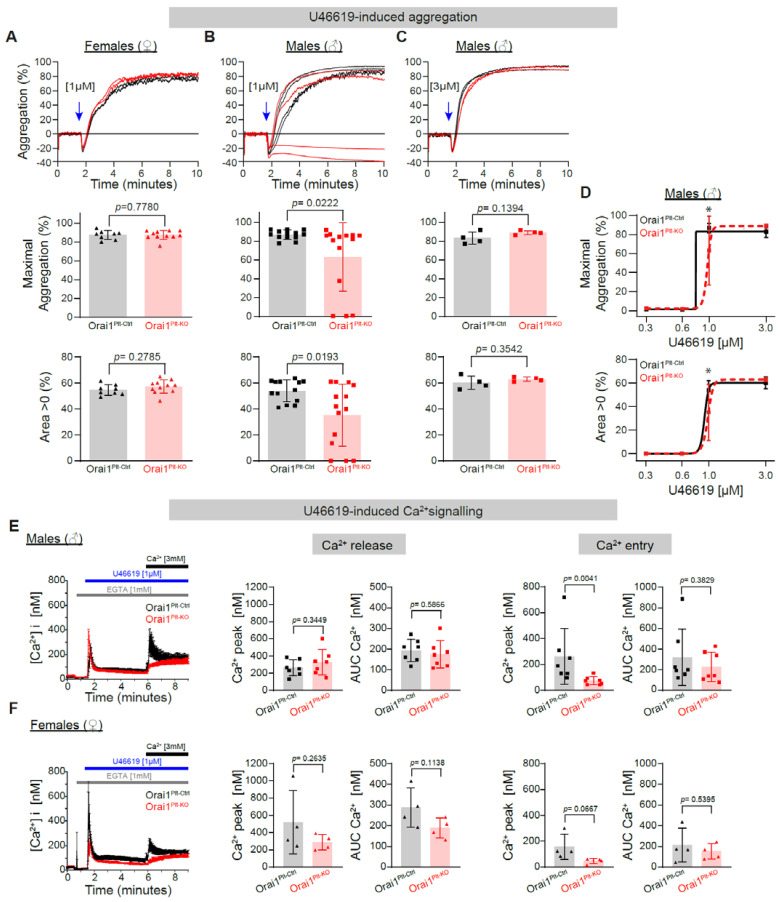
Impaired Thromboxane A2 analogue U46619-induced platelet aggregation and Ca^2+^ signalling in Orai1-deficient platelets from male mice. U46619-induced platelet aggregation in washed platelets from Orai1^Plt-Ctrl^ (black) and Orai1^Plt-KO^ (red) mice was analysed (**A**–**D**). Representative original aggregation traces (**upper panels**), analysis of maximal aggregation (**middle panels**) and area under the aggregation curve (**lower panels**) after stimulation with 1 µM U46619 of platelets from female (**A**) or male (**B**) mice; similarly, analysis of platelets from male mice stimulated with 3 µM U46619 are depicted (**C**). Dose-response curves for the maximal aggregation (**upper panel**) and the area under the aggregation curve (**lower panel**) after U46619 stimulation of male platelets are shown (**D**). Each dot in (**A**–**C**) corresponds to one independent platelet preparation. In (**D**) numbers of independent platelet preparations per U49916 concentration were: 0.3 µM = 4, 0.6 µM = 4, 1 µM = 13/14 and 3 µM = 4. Blue arrows indicate the agonist stimulation time point. In (**E**,**F**) U46619-induced Ca^2+^ signalling was analysed in platelets from male (**E**) (*n* = 7) and female mice (**F**) (*n* = 4). Average Ca^2+^ traces of Fura-2 LR/AM loaded platelets (**left panels**) after U46619 stimulation are shown; corresponding statistical analysis of maximal Ca^2+^ peaks, and area under the curve (AUC) of Ca^2+^ release (**middle panels**) and Ca^2+^ entry (**right panels**) are depicted. Error bars indicate SD. Original Ca^2+^ traces are represented as mean ± SEM for better visualization. *p*-values or * *p* < 0.05 were calculated according to the unpaired Student’s *t*-test (Panels **A** −Area > 0−, **C** and **E** −Ca^2+^ Release− and **F**) or Mann-Whitney U test (Panels **A** −max. aggregation, **B** and **E** −Ca^2+^ entry−) after Shapiro-Wilk normality test.

**Figure 3 cells-11-03225-f003:**
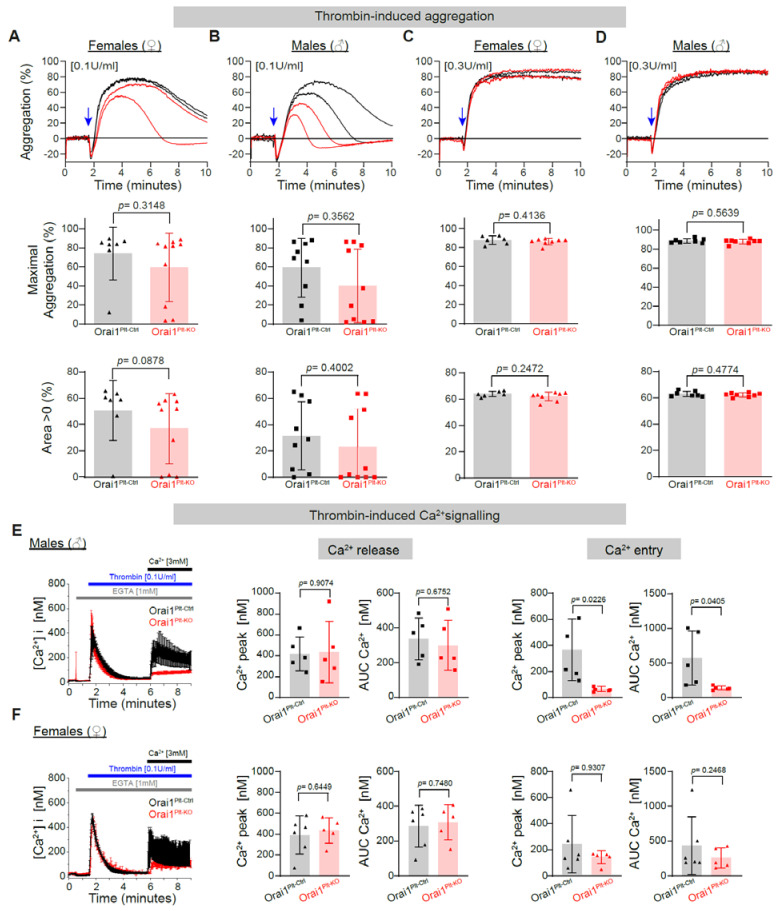
Preserved thrombin-induced platelet aggregation in Orai1 deficient platelets but impaired thrombin-induced Ca^2+^-entry in Orai1 deficient platelets from male mice. Thrombin-induced platelet aggregation in washed platelets from Orai1^Plt-Ctrl^ (black) and Orai1^Plt-KO^ (red) mice was analysed (**A**–**D**). Representative original aggregation traces (**upper panels**), analysis of maximal aggregation (**middle panels**) and area under the aggregation curve (**lower panels**) after stimulation with 0.1 U/mL (**A**,**B**) or 0.3 U/mL (**C**,**D**) thrombin of platelets from female (**A**,**C**) or male (**B**,**D**) mice. Blue arrows indicate the agonist stimulation time point. In (**E**,**F**) thrombin-induced Ca^2+^ signalling was analysed in platelets from male (**E**) (*n* = 5) and female mice (**F**) (*n* = 6). Average Ca^2+^ traces of Fura-2 LR/AM loaded platelets (**left panels**) after thrombin stimulation are shown; corresponding statistical analysis of maximal Ca^2+^ peaks and area under the curve (AUC) of Ca^2+^ release (**middle panels**) and Ca^2+^ entry (**right panels**) are depicted. Error bars indicate SD. Original Ca^2+^ traces are represented as mean ± SEM for better visualization. *p*-values were calculated according to the unpaired Student’s *t*-test (Panels **C**–Area > 0 of thrombin 0.3U/mL–, **D**, **E** and **F**–Ca^2+^ release–) or Mann-Whitney U test (Panels **A**, **B** and **C**–not Area > 0 of thrombin 0.3 U/mL– and **F**–Ca^2+^ entry–) after Shapiro-Wilk normality test.

**Figure 4 cells-11-03225-f004:**
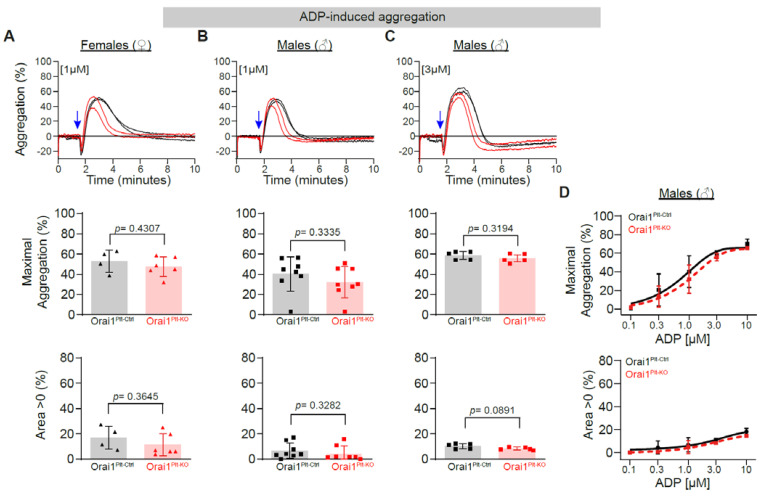
Preserved ADP-induced platelet aggregation in Orai1 deficient platelets. ADP-induced platelet aggregation in washed platelets from Orai1^Plt-Ctrl^ (black) and Orai1^Plt-KO^ (red) mice was analysed (**A**–**D**). Representative original aggregation traces (**upper panels**), analysis of maximal aggregation (**middle panels**) and area under the aggregation curve (**lower panels**) after stimulation with 1 µM ADP of platelets from female (**A**) or male (**B**) mice; similarly, analysis in platelets from male mice stimulated with 3 µM ADP are depicted (**C**). Dose-response curves for the maximal aggregation (**upper panel**) and the area under the aggregation curve (**lower panel**) after ADP stimulation of male platelets are shown (**D**). Each dot in (**A**–**C**) corresponds to one independent platelet preparation. In (**D**) numbers of independent platelet preparations per ADP concentration were: 0.1 µM = 3, 0.3 µM = 5, 1 µM = 8, 3 µM = 5 and 10 µM = 3. Blue arrows indicate the agonist stimulation time point. Error bars indicate SD. *p*-values were calculated according to the unpaired Student’s *t*-test (Panels **A**, **B** and **C** −except for Area > 0 of ADP 1 µM from male mice−) or Mann-Whitney U test (Area > 0 of ADP 1 µM from male mice) after Shapiro-Wilk normality test.

**Figure 5 cells-11-03225-f005:**
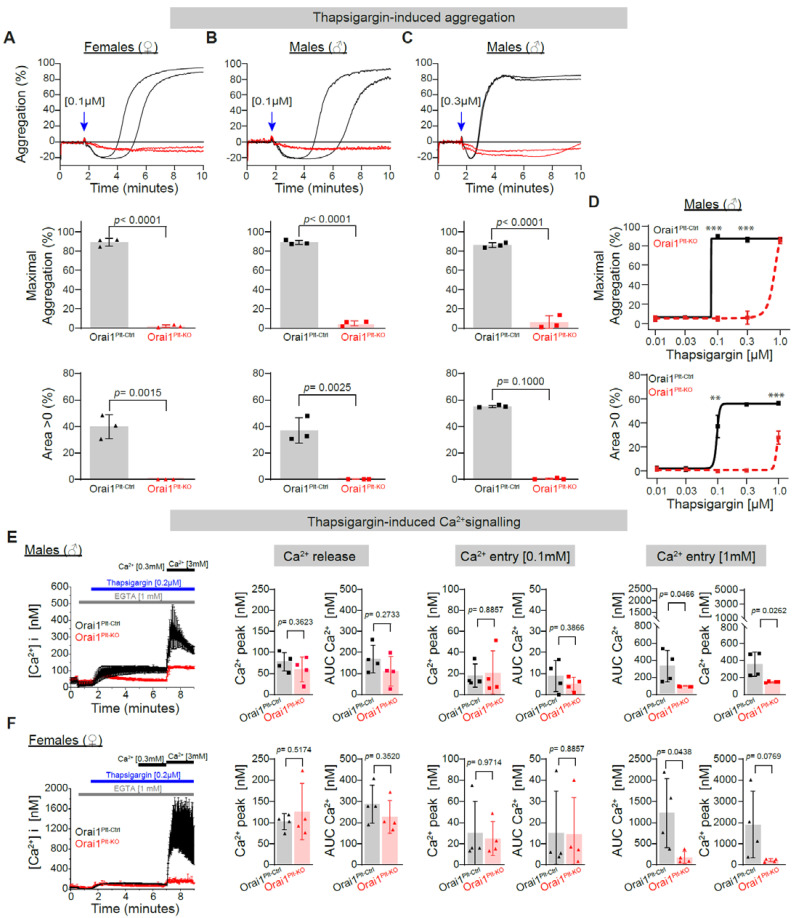
Abolished thapsigargin-induced platelet aggregation and Ca^2+^ entry in Orai1 deficient platelets. Thapsigargin-induced platelet aggregation in washed platelets from Orai1^Plt-Ctrl^ (black) and Orai1^Plt-KO^ (red) mice was analysed (**A**–**D**). Representative original aggregation traces (**upper panels**), analysis of maximal aggregation (**middle panels**) and area under the aggregation curve (**lower panels**) after stimulation with 0.1 µM Thapsigargin of platelets from female (**A**) or male (**B**) mice; similarly, analysis in platelets from male mice stimulated with 0.3 µM thapsigargin are depicted (**C**). Dose-response curves for the maximal aggregation (**upper panel**) and the area under the aggregation curve (**lower panel**) after thapsigargin stimulation of male platelets are shown (**D**). Each dot in (**A**–**C**) corresponds to one independent platelet preparation. In (**D**) numbers of independent platelet preparations per thapsigargin concentration were: 0.01 µM = 3, 0.03 µM = 3, 0.1 µM = 3, 0.3 µM = 3 and 1 µM = 4. Blue arrows indicate the agonist stimulation time point. In (**E**,**F**) thapsigargin-induced (0.2 µM) Ca^2+^ signalling was analysed in platelets from male (**E**) (*n* = 4) and female mice (**F**) (*n* = 4). Average Ca^2+^ traces of Fura-2 LR/AM loaded cells (**left panels**) are depicted; corresponding statistical analysis of maximal Ca^2+^ peaks and area under the curve (AUC) of Ca^2+^ release and Ca^2+^ entry (under two different external Ca^2+^ concentrations: 0.1 mM and 1 mM) are shown. Error bars indicate SD. Original Ca^2+^ traces are represented as mean ± SEM for better visualization. *p*-values or ** *p* < 0.01 and *** *p* < 0.001 were calculated according to the unpaired Student’s *t*-test (Panels **A**, **B** and **C** -except Area > 0 of TG 0.3 µM-, **E** -except for Ca^2+^ peak after addition of 0.1 mM- and **F** −except for Ca^2+^ entry after 0.1 mM−) or Mann-Whitney U test (Panels **C** −Area > 0 of TG 0.3 µM−, **E** −Ca^2+^ peak after addition of 0.1 mM Ca^2+^− and **F** −Ca^2+^ entry after 0.1 mM−) after Shapiro-Wilk normality test.

## Data Availability

Not applicable.

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
