# Peer review of "Reduction in SOCE and Associated Aggregation in Platelets from Mice with Platelet-Specific Deletion of Orai1"

_cells, 2022, doi:10.3390/cells11203225_

Round 1

Reviewer 1 Report

In this paper Linlin Yang and collegues performed numerous experiments on wild type mice and mice with and Orai1-deficient platelets to demonstrate the role of store-operated calcium entry in platelet activation. The authors performed two types of experiments - washed platelet aggregation and measurement of average calcium signal in platelet suspention. Altogether, the conclusions of the paper are based on the data. However, experiments in platelet suspentions often give incorrect results due to unaccouned for secondary platelet activation. Also, there is a number of technical questions those require clarification before the paper could be accepted for publication.

Major comments

  1. All experiments were performed in platelet suspention, therefore, secondary platelet activation occurs in most cases. Either single cell, flow cytometry or experiments with inhibition of secondary activation should be performed to support the conclusions. Especially, for extracellular calcium entry the ATP-induced P2X1-mediated process is of interest.
  2. In the calcium measurement experiments 1 mM calcium was added to 1 mM EGTA, which results in approx 100 uM of calcium in the extracellular space (depending on pH, temperature, ionic strength of the solution, etc.) - in any case it is much lower that the physiological 1.3 mM of free calcium. On the contrary, 2 mM of calcium was used in platelet aggregation experiments. It is necessary to add a discussion about the correctness of comparing the behavior of platelets at such different calcium concentrations. 
  3. In a large number of studies, it was previously shown that in platelets, SOCE primarily affects the formation of a procoagulant subpopulation. Given the fact that in this work there is no possibility of observing platelet subpopulations, could the observed results be explained by the inhibition of procoagulant platelet formation?

Minor comments

  1. Lines 18-21. The presence of extracellular calcium and its concentration should be mentioned here, as well as the methods for calcium entry and release measurements.
  2. Line 43. This statement is misleading, as ROCE in platelets is considered to be either IP3-dependent (through IP3R-1 on platelet plasma membrane) or ATP-dependent through P2X1. The functioning of DAG-dependent TRPC channel was not sufficiently proved. Besides, it is well known that platelet calcium signalling is mainly dependent on IP3-induced calcium release from DTS, which should be mentioned here to avoid ambiguity.  
  3. Lines 49-50. This statement is controversial and needs additional references to the literature
  4. Line 57. This statement is not correct, as IP3 does not directly activate SOCE
  5. Line 61, 65, 304, 343. Mice and humans do not have thrombocytes, only platelets.
  6. Lines 63-64. This statement is controversial, as STIM proteins could not leave the inner membrane.
  7. Line 125. Inharmonious sentence
  8. Line 142. As given in ref 40, Kd for fura-2 equals 160 nM for 20C and 224 nM for 37C, why was the value 290 nM used?
  9. Which statistical test was applied to which experimental dataset should be clearly indicated in the figure legends.
  10. Discussion could be imroved by additional consideration of the results obtained for human platelets in the absence of extracellular calcium.

Author Response

We are thankful to the review for the careful reading of our manuscript and the contribution made by the raised questions, which we are addressed in the attached pdf file.

Reviewer 2 Report

Calcium signalling in platelets through store operated Ca2+ entry (SOCE) or receptor-operated Ca2+ entry (ROCE) mechanisms is crucial for platelet activation and function. Orai1 proteins have been implicated in platelet’s SOCE.

Based on this theoretical knowledge, the authors evaluated the contribution of Orai1 proteins to these processes.

SOCE and aggregation induced by Thapsigargin up to a concentration of 0.3 µM was abrogated in Orai1-deficient platelets. Receptor-operated Ca2+ entry and platelet aggregation induced by U46619 or thrombin were partially affected by Orai1 deletion depending on the gender. Thus, the presented results reinforce the indispensability of Orai1 proteins for SOCE in murine platelets.

The manuscript is written on nineteen pages, three of which are references and further two contain supplementary material. It contains a lot of illustrative file material including five figures.

It is written on the high scientific level with an available description of the method used in the study to enable the correct repetition of the experiment.

Data were analysed and processed with Excel 2010 (Microsoft), Graphpad Prism 8.0 and OriginPro 2019G ver. 9.6.0.172 (OriginLab). Values are shown as mean ± standard deviation (SD), except for original traces of Ca2+ imaging. For dose-response curves a non-linear regression was done for graphical representation. Statistical significances were assumed when p<0.05. Statistical analysis was performed using the appropriate tests (unpaired Student’s t-test or Mann-Whitney U test) after normality test (Shapiro-Wilk normality test).

The authors conclude that in the physiological context activation of relevant pathways in platelets by agonist such as collagen, thromboxane A2, thrombin or ADP triggers downstream parallel activation of PLC or PLC2 isoforms; PLC activation is capable to recruit both SOCE and ROCE dependent Ca2+ entry mechanisms, by further IP3 and diacylglycerol formation and or PIP2 depletion. Ca2+-entry and platelet aggregation induced by U46619 or thrombin were partially affected by Orai1 deletion depending on the gender. ADP, collagen and CRP aggregation from Orai1- deficient platelets was comparable with control cells, as well as CRP-induced Ca2+-signalling.

The presented results support the concept of Orai1 involvement into SOCE mechanism in murine platelets, thus contributing to understanding of its role in agonist-dependent signalling and pointing out the importance to analyse separately platelets from both genders.

Content suggestions:

  1. Why the authors did not use for the induction of aggregation also arachidonic acid or epinephrine ?

  2. How can the authors explain the difference in results with regards to the gender of mice used in the study ?

Author Response

(The authors gave the same response as above.)

Round 2

Reviewer 1 Report

The authors thoroughly and fully answered all my questions, I have no more comments